# Unifying Top-down and Bottom-up Recurrent Attention Model for Visual Recognition

## Abstract

The idea of using the recurrent neural network for visual attention has gained popularity in computer vision community. Although the recurrent attention model (RAM) leverages the glimpses with more large patch size to increasing its scope, it may result in high variance and instability. For example, we need the Gaussian policy with high variance to explore object of interests in a large image, which may cause randomized search and unstable learning. In this paper, we propose to unify the top-down and bottom-up attention together for recurrent visual attention. Our model exploits the image pyramids and Q-learning to select regions of interests in the top-down attention mechanism, which in turn to guide the policy search in the bottom-up approach. In addition, we add another two constraints over the bottom-up recurrent neural networks for better exploration. We train our model in an end-to-end reinforcement learning framework, and evaluate our method on visual classification tasks. The experimental results outperform convolutional neural networks (CNNs) baseline and the bottom-up recurrent attention models on visual classification tasks.

## 1 Introduction

Recurrent visual attention model Mnih et al. (2014) (abbreviated as RAM) leverages reinforcement learning Sutton & Barto (1998) and recurrent neural networks Schuster et al. (1997); LeCun et al. (2015) to recognize the objects of interests in a sequential manner. Specifically, RAM models object recognition as a sequential decision problem based on reinforcement learning. The agent can adaptively select a series of regions based on its state and make decision to maximize the expected return. The advantages of RAM are follows: firstly, its computation time scales linearly with the complexity of the patch size and the glimpse length rather than the whole image, through a sequence of glimpses and interactions locally. Secondly, this model is very flexible and effective. For example, it considers both local and global features by using more number of glimpses, scales and large patch sizes to enhance its performance.

Unfortunately, the idea of using glimpse in the local region limits its understanding to the holistic view and restricts its power to capture the semantic regions in the image. For example, its initial location and patch extraction is totally random, which will take more steps to localize the target objects. Although we can increase the patch size to cover larger region, it will increase the computation time correspondingly. In the same time, we need to set a high variance (if we use Gaussian policy) to explore unseen regions. Thinking a situation that you were in desert and you were looking for water, but you did not have GPS/Map over the whole scene. What would you do to find water? You might need to search each direction (east, south, north, and west), that is random searching and time-consuming. In this paper, we want to provide a global view to localize where to search. For example, the process of human perception is to scan the whole scene, and then may be attracted by specific regions of interest. In a high level view, it can help us to ignore non-important parts and focus information on where to look, and further to guide us to different fixations for better understanding the internal representation of the scene.

To solve 'where to look' using reinforcement learning, the desired algorithm needs to satisfy the following criterions: (1) it is stable and effective; (2) it is computationally efficient; (3) it must be more goal-oriented with global information; (4) instead of random searching, it attends regions of interest even over large image; (5) it has better exploration strategy on 'where to look'. For a large

image, the weakness of RAM is exposed throughly, because it relies on local fixations to search the next location, which increasing the policy uncertainty as well as the learning time to optimizing the best in the interaction sequence. Inspired by the hierarchical image pyramids Adelson et al. (1984); Lin et al. (2017) and reinforcement learning Watkins & Dayan (1992); Kulkarni et al. (2016), we propose to unify the top-down and bottom-up mechanism to overcome the weakness of recurrent visual attention model.

In the high-level, we take a top-down mechanism to extract information at multiple scales and levels of abstraction, and learn to where to attend regions of interests via reinforcement learning. While in the low-level, we use the similar recurrent visual attention model to localize objects. In particular, we add another two constraints over the bottom-up recurrent neural networks for better exploration. Specifically, we add the entropy of image patch in the trajectory to enhance the policy search. In addition, we constrain the region that attend to should be more related to target objects. By combining the sequential information together, we can understand the big picture and make better decision while interacting with the environment.

We train our model in an end-to-end reinforcement learning framework, and evaluate our method on visual classification tasks including MNIST, CIFAR 10 classes dataset and street view house number (SVHN) dataset. The experimental results outperform convolutional neural networks (CNNs) baseline and the bottom-up recurrent attention models (RAM).

## 2 RELATED WORK

Since the recurrent models with visual attention (RAM) was proposed, many works have been inspired to either use RNNs or reinforcement learning to improve performance on computer vision problems, such as object recognition, location and question answering problems. One direction is the top-down attention mechanism. For example, Caicedo and Lazebnik extends the recurrent attention model and designs an agent with actions to deform a bounding box to determine the most specific location of target objects Caicedo & Lazebnik (2015). Xu et al. (2015) introduces an attention based model to generate words over an given image. Basically, it partitions image into regions, and then models the importance of the attention locations as latent variables, that can be automatically learned to attend to the salient regions of image for corresponding words. Wang et al. Wang et al. (2018) leverages the hierarchical reinforcement learning to capture multiple fine-grained actions in sub-goals and shows promising results on video captioning.

Another trend is to build the interpretation of the scene from bottom-up. Butko & Movellan (2009) builds a Bayesian model to decide the location to attend while interacting with local patches in a sequential process. Its idea is based on reinforcement learning, such as partially observed Markov decision processes (POMDP) to integrate patches into high-level understanding of the whole image. The design of RAM Mnih et al. (2014) takes a similar approach as Butko & Movellan (2008; 2009), where reinforcement learning is used to learn 'where to attend'. Specifically, at each point, the agent only senses locally with limited bandwidth, not observe the full environment. Compared Butko & Movellan (2009), RAM leverages RNNs with hidden states to summarize observations for better understanding the environment. Similarly, drl-RPN Pirinen & Sminchisescu (2018) proposes a sequential attention model for object detection, which generates object proposal using deep reinforcement learning, as well as automatically determines when to stop the search process.

Other related work combine local and global information to improve performance on computer vision problems. Attention to Context Convolution Neural Network (AC-CNN) exploits both global and local contextual information and incorporates them effectively into the region-based CNN to improve object detection performance Li et al. (2017). Anderson et al. Anderson et al. (2018) proposes to combine bottom-up and top-down attention mechanism for image captioning and visual question answering. They uses the recurrent neural networks (specifically, LSTM) to predict an attention distribution over image regions from bottom-up. However, no reinforcement learning technique used to adaptively focus objects and other salient image regions of interests.

RAM learns to attend local regions while interacting with environment in a sequential decision process with the purpose to maximum the cumulative return. However, if the scope of local glimpse is small, then we lose the high level context information in the image when making decision in an sequential manner. In contrast, if we increase the patch size to the scale as large as the original

image, then we lose the attention mechanism, as well as its efficiency of this model. Moreover, when we increase the scale with the fixed patch size, we will deprive distinguished features after resizing back to original patch size. Also it is a challenge to balance the Gaussian policy variance and patch size. Since the initial location is randomly generated, we need to set a large variance in Gaussian policy to increase its policy exploration. However, it will result in high instability in the learning process. In this paper, we propose to unify bottom-up and top-down mechanism to address these issues mentioned above. In addition, we introduce entropy into policy gradient in order to attend regions with high information content.

## 3 BACKGROUND

### 3.1 REINFORCEMENT LEARNING

The objective of reinforcement learning is to maximize a cumulative return with sequential interactions between an agent and its environment Sutton & Barto (1998). At every time step $t$, the agent selects an action $a_t$ in the state $s_t$ according its policy and receives a scalar reward $r_t(s_t, a_t)$, and then transit to the next state $s_{t+1}$ with probability $p(s_{t+1}|s_t, a_t)$. We model the agent's behavior with $\pi_\theta(a|s)$, which is a parametric distribution from a neural network.

Suppose we have the finite trajectory length while the agent interacting with the environment. The return under the policy $\pi$ for a trajectory $\tau = (s_t, a_t)_{t=0}^T$

$$J(\theta) = E_{\tau \sim \pi_\theta(\tau)}[\sum_{t=0}^{T} \gamma^t r(s_t, a_t)] = E_{\tau \sim \pi_\theta(\tau)}[R_0^T] \tag{1}$$

where $\gamma$ is the return discount factor, which is necessary to decay the future rewards ensuring bounded returns. $\pi_\theta(\tau)$ denotes the distribution of trajectories below

$$\rho(\tau) = \pi(s_0, a_0, s_1, ..., s_T, a_T)$$
$$= p(s_0) \prod_{t=0}^{T} \pi_\theta(a_t|s_t) p(s_{t+1}|s_t, a_t) \tag{2}$$

The goal of reinforcement learning is to learn a policy $\pi$ which can maximize the expected returns.

$$\theta = \arg\max J(\theta) = \arg\max E_{\tau \sim \pi_\theta(\tau)}[R_0^T] \tag{3}$$

**Policy gradient**: Take the derivative w.r.t. $\theta$

$$\nabla_\theta J(\theta) = \nabla_\theta E_{\tau \sim \pi_\theta(\tau)}[R_0^T] = \nabla_\theta \int \rho(\tau) R_0^T d\tau$$
$$= \int \nabla_\theta \rho(\tau) R_0^T d\tau = \int \rho(\tau) \frac{\nabla_\theta \rho(\tau)}{\rho(\tau)} R_0^T d\tau$$
$$= E_{\tau \sim \pi_\theta(\tau)}[\nabla_\theta \log \pi_\theta(\tau) R_0^T]$$
$$= E_{\tau \sim \pi_\theta(\tau)}[\nabla_\theta \log \pi_\theta(\tau)(R_0^T - b)] \tag{4}$$

**Q-learning**: The action-value function describes what the expected return of the agent is in state $s$ and action $a$ under the policy $\pi$. The advantage of action value function is to make actions explicit, so we can select actions even in the model-free environment. After taking an action $a_t$ in state $s_t$ and thereafter following policy $\pi$, the action value function is formatted as:

$$Q^\pi(s_t, a_t) = \mathbb{E}_{\tau \sim \pi_\theta(\tau)}[R_t|s_t, a_t]$$
$$= \mathbb{E}_{\tau \sim \pi_\theta(\tau)}[\sum_{i=t}^{T} \gamma^{(i-t)} r(s_i, a_i)|s_t, a_t] \tag{5}$$

To get the optimal value function, we can use the maximum over actions, denoted as $Q^*(s_t, a_t) = \max_\pi Q^\pi(s_t, a_t)$, and the corresponding optimal policy $\pi$ can be easily derived by $\pi^*(s) \in \arg\max_{a_t} Q^*(s_t, a_t)$.

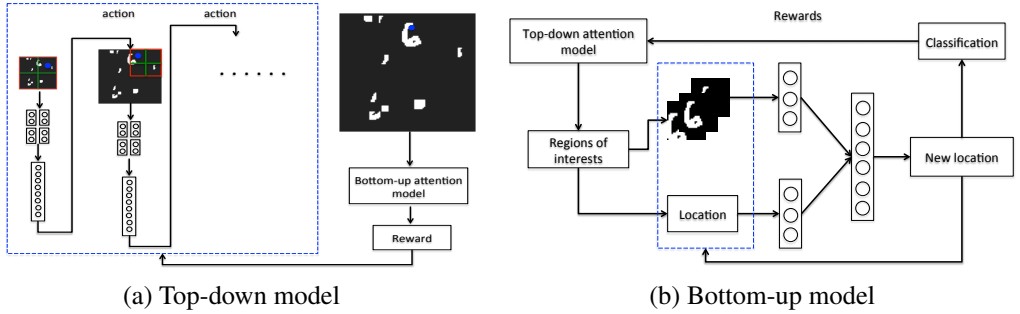

(a) Top-down model                    (b) Bottom-up model

Figure 1:   (a) The top-down attention model: it divides the region (marked as red) into 4 sub-regions in the current resolution, and then concatenates features of each region (extracted from neural networks) as in the input to action network to decide which sub-region it should choose to attend; This process repeats to select sub-region in the next level of image pyramid (b) the bottom-up attention model: based on the location output from the top-down attention model, it extracts features at different scales at the current location and then combines location feature as input to recurrent neural networks and action network to decide next location.

## 3.2   RECURRENT ATTENTION MODEL

Instead of processing the whole image $X$, recurrent attention model (RAM) Mnih et al. (2014) uses the recurrent neural networks (RNN) to guide the agent to make a sequential decision process while interacting with a visual environment. At the time step $t$, the agent is at location $L_t$, observing image patch $x_t$ with different patch size, and then it makes decision to next location $L_{t+1}$. Finally, the agent executes action and receives a scalar reward (which will be used to correct its decision). In classification task, the agent will get positive reward if it predicts right. The goal of the agent is to maximize the total sum of such rewards.

Since RAM extracts one patch at each time step, the number of pixels it processes in the glimpse $x_t$ is much smaller than that in the original image $X$. Compared to deep neural networks, it is much efficient, i.e. the computational cost of a single glimpse is independent of the size of the image. The model structure of RAM is listed below with paraphrase:

Glimpse network $f_g$: it extracts features around position $L_t$ at different scales via its sensor, and then combines visual features and position features together via glimpse network $f_g$ to generate the glimpse feature vector $G_t$.

RNN $f_h$: it takes the glimpse vector $G_t$ and $H_{t-1}$ to output the next the hidden state $H_t$, which summarizes its history knowledge over the environment.

Action network $f_l$: it uses the output from $f_h$, and then decides the agent next location $L_t$ based on the internal hidden state $H_t$.

Classification network $f_y$: it integrates all information to the current state in the sequence and decides the class of the sequence or image (which further decides the reward the agent will receive).

## 4   UNIFIED VISUAL ATTENTION MODEL

In this section, we present an unified framework, which combines top-down and bottom-up mechanism for visual attention.

### 4.1   TOP-DOWN ATTENTION MODEL

The top-down attention mechanism is implemented to attend significant regions by following coarse-to-fine image pyramid. In the coarse level, we divide the image into regions, and then select the region based on its importance. Further, we map the region into the next level pyramid with higher resolution and divide it into subregions and repeat the process until to the final (original image). In Figure 1(a), we show an example image, which is divided into $2 \times 2$ regions in the coarsest level.

In the next level, each region repeats this like quad tree image representation. How to select one from four regions is based on the reward backups from the bottom-up attention model. Since we divide the current level image into $2 \times 2$ regions, we need a strategy to select one region to look, which tries to solve "where to look" in our top-down attention model.

The action that learns which region to look is based on Q-learning, which is parametrized by 2-layer networks in our work. As for the agent, the state is the image (representation) at the current level $\ell$, and its actions are to choose one of its 4 regions, which can map to subregions in the level $\ell + 1$. The reward is from the bottom-up model, which will be introduced in the next part. For the object recognition problem, if we classify the image correctly from bottom-up mechanism, then we get positive reward, otherwise negative reward backward to top-down attention model.

As for the network structure, we can use full connected neural networks (or CNN) to extract a set of features from the image in each level in the hierarchy. For instance, if we partition the low resolution image into $\mathcal{A}$ regions (where we use $\mathcal{A}$ to denote the action space), then we get the $|\mathcal{A}|$ vectors, each of which represents as $N$ dimensional vector corresponding to a part of the image in that level

$$\ell = \{\mathbf{v}_1, \mathbf{v}_2, ..., \mathbf{v}_{|\mathcal{A}|}\}, \mathbf{v}_i \in \mathbb{R}^N \tag{6}$$

where each region is represented as $N$-dimensional vector $\mathbf{v}_i$. Then we concatenate all vectors together to form the (sub)image representations at the current level

$$s_\ell = [\mathbf{v}_1, ..., \mathbf{v}_{|\mathcal{A}|}], s_\ell \in \mathbb{R}^{|\mathcal{A}|N} \tag{7}$$

To select one region from the total $\mathcal{A}$ regions in each level, we leverage Q-learning, which as an off-policy RL algorithm and has been extensively studied since it was proposed Watkins & Dayan (1992). The Q-value function $q(s, a)$ should output $|\mathcal{A}|$-dimensional values to indicate the importances of the $\mathcal{A}$-regions. correspondently. Thus, we add another layer of full connected networks which outputs $|\mathcal{A}|$-dimensional values to decide the probability (softmax) to decide the next region to look at in the next level. In our case, the agent needs to select regions according to its q-value.

Suppose we use neural network parametrized by $\theta_q$ to approximate Q-value in the interactive environment. To update Q-value function, we minimize the follow loss:

$$y_\ell = r(s_\ell, a_\ell) + \gamma \max_{a_{\ell+1}} Q(s_{\ell+1}, a_{\ell+1}; \theta_q) \tag{8}$$

$$L(\theta_q) = \mathbb{E}_{s_\ell \sim p_\pi, a_\ell \sim \pi}[(Q(s_\ell, a_\ell; \theta_q) - y_\ell)^2] \tag{9}$$

where $s_{\ell+1}$ is the state (or image region) in the level $\ell + 1$, $a_{\ell+1}$ is the action to pick one region from the total $\mathcal{A}$ regions. $y_\ell$ is from Bellman equation, and we update it from its action $a_{t+1} \in \{1, 2, .., |\mathcal{A}|\}$ which is taken from frozen policy network (actor) to stabilizing the learning. Since we don not know which region to focus until the last layer reward from the bottom-up model, so we design $r(s_\ell, a_\ell) = 1$ if $\ell$ is the last pyramid layer (original image) as well as we recognize object correctly from bottom-up model, otherwise $r(s_\ell, a_\ell) = 0$. $Q(s_\ell, a_\ell; \theta_q)$ is approximated with a two-layer neural network (one convolution layer followed with fully connected layer).

In the current image level $\ell$, the agent takes step to select one region to focus on according to q-value, and then further repeat to select subregions based on q-value in the next level $\ell + 1$. How to learn the q-value function will be dependent on the final reward in the bottom-up model in the next section. We can minimize Eq. 9 to learn parameter $\theta_q$.

## 4.2 BOTTOM-UP ATTENTION MECHANISM

Our bottom-up attention model takes the similar approach as RAM Mnih et al. (2014), to use the recurrent neural networks (RNN) to guide the agent to make a sequential decision process while interacting with a visual environment.

At each location, the agent observes the low-level scene properties in a local region and build up bottom-up processes over time to integrate information from visual environment in order to determine how to act and how to deploy its sensor most effectively.

$$G_t = f_g(x_t, L_t; \theta_g) \tag{10a}$$

$$H_t = f_h(H_{t-1}, G_t; \theta_h) \tag{10b}$$

$$L_t = f_l(H_t; \theta_l) \tag{10c}$$

$$\boldsymbol{\alpha}_t = f_y(H_t; \theta_y) \tag{10d}$$

where $x_t$ is the local patch or observation of the visual environment (image), with location specified by $L_t$. $f_g$ combines visual feature $G_{img}(x_t)$ and location feature $G_{loc}(L_t)$ together, where visual feature $G_{img}(x_t)$ consists of three convolutional hidden layers with normalization and pooling layers followed by a fully connected layer, and $G_{loc}(L_t)$ is from full connected hidden layer. And we use $G_t = G_{img}(x_t) + G_{loc}(L_t)$ to get glimpse feature at $L_t$. $H_t$ integrates the previous hidden state and $G_t$ to generate next hidden state, in order to capture the whole sequential information. $f_y$ is the classifier, which integrates all information to the current state in the sequence and decides the class of the image (which further decides the reward the agent will receive). it is formulated using a softmax output and for dynamic environments, its exact formulation depends on the action set defined for that particular environment. For classification with $K$ classes, we have $\hat{y} = \arg\max_{k \in K} \boldsymbol{\alpha}_{tk}, \boldsymbol{\alpha}_t \in \mathbb{R}^{1 \times K}$.

Compared to RAM, we add anther two constraints which balances the trade-off between exploration and exploitation. On the one hand, we want to take action to select high entropy regions to better explore the whole environment. On the other hand, we hope the regions selected have high confidence on classification task. To sum up, our contributions are followings:

(1) Context constraint: the basic idea is that for the regions selected in policy search, we should assign them more weights. $\boldsymbol{\alpha}_t$ is the softmax output at each step $t$, which in fact is the weight to measure the importance of the current hidden state, or whether we should attend to current location. Once the weight $\boldsymbol{\alpha}_t$ (which sums to one) are computed, then the global context vector $C$ and its softmax output $\hat{z}$ is computed as

$$C = \sum_{t=1}^{T} \boldsymbol{\alpha}_{ty} H_t$$
$$\hat{z} = f_c(C) \tag{11}$$

where $f_c$ is the two-layer networks, with a fully connected layer and a softmax output layer. We hope $\hat{z}$ matches its groundtruth $\hat{z} = y$.

(2) Entropy constraint with better exploration: Entropy has been widely use to measure information, where a higher entropy value indicates a relatively richer information content. As a bottom-up algorithm, we want our model gives more weight on regions with higher entropy. Specifically, we want to skip all black or white regions while searching objects. Thus, for each region we extracted, we threshold it and binarialize it into 2 bins and then compute its entropy. In order to take action towards high entropy regions, we modify the reward function as

$$J(\theta) = E_{\tau \sim \pi_\theta(\tau)} \log \pi_\theta(\tau)(entropy(x_t) + \lambda)(R_0^T - b) \tag{12}$$

where $entropy(x_t)$ is the entropy over the patch located by $L_t$, $\lambda$ is the constant to scale the importance at the current time step, $R_0$ is the reward over sequence and $b$ is the baseline which can be approximated by neural network. Since the action to localize the next region is backward from reward, then we decide reward $R_0$ as: if the model makes the right decision, then $R_0 = 1$, otherwise $R_0 = 0$. The location network $f_l$ decides action at $L_t$, and the policy gradient can be formalized as

$$\nabla_\theta J(\theta) = E_{\tau \sim \pi_\theta(\tau)}[\nabla_\theta \log \pi_\theta(\tau)(entropy(x_t) + \lambda)(R_0^T - b)] \tag{13}$$

To learn action in $L_t$, we use Gaussian policy with variance $\sigma$ which measures the exploration scale.

The final objective function in the bottom-up algorithm is hybrid supervised loss as follow:

$$loss = -y\log\alpha_T - \beta_1 y\log\hat{z} + \beta_2 J(\theta) \tag{14}$$

where $\beta_1$ and $\beta_2$ are the weights to balance the importance of each term.

### 4.3 ALGORITHM

We outline the sketch of our training process in Algorithm 1. Note that we use another frozen Q-learning network to stabilize the learning procedure.

---

**Algorithm 1** Unified visual attention model

---
1: Initialize model parameters in Eq. 9 and Eq. 10,
2: **for** each $epoch$ **do**
3:     **for** each $image, label(y)$ **do**
4:         Decide the action $a_i = \max_{a_\ell} Q(s_\ell, a_\ell)$;
5:         Get the corresponding region $\mathbf{v}_i$ via $a_i \in \mathcal{A}$;
6:         Randomly sample location $L_0$ from image region $\mathbf{v}_i$;
7:         Randomly generate initial hidden state $H_0$;
8:         **for** each glimpse $t = 0$ to $T$ **do**
9:             Run the RNN model to decide next location to focus via Eq. 10a, 10b and 10c;
10:             **if** $condition$ is ok **then**
11:                 Run classifier via Eq. 10d;
12:                 $\hat{y}_t = \arg\max_{k \in K} \boldsymbol{\alpha}_{tk}$;
13:             **end if**
14:         **end for**
15:         Evaluate the sequential reward $R = [y == \hat{y}]$
16:         **if** $R$ is correct in the whole sequence **then**
17:             $r(s_\ell, a_\ell) = 1$;
18:         **else**
19:             $r(s_\ell, a_\ell) = 0$;
20:         **end if**
21:         Update Q-learning (top-down) model parameter $\theta_q$ by minimizing loss in Eq. 9;
22:         Update bottom-up model parameters $\{\theta_g, \theta_h, \theta_l, \theta_y\}$ by minimizing loss in Eq. 14;
23:     **end for**
24: **end for**

---

## 5 EXPERIMENTS

We evaluated our approach on MNIST, CIFAR10 and SVHN datasets. As for parameter setting, we use 2-level pyramid representation in the top-down q-learning: the coarse and the original image level. Basically, the coarse level image is the half resolution (size) of the original image, and we divide the coarse image into 2 by 2 subregions, then we can learn the location in the coarse level, which in turn can be mapped back to the original image. In the bottom-up attention model, we use Adam algorithm with learning rate $3e - 4$, batch size 128, $\lambda = 0.5$, $\beta_1 = 1$, $\beta_2 = 0.01$ and $\sigma = 0.15$. Without other specification, we use the same parameter setting above in all the following experiments.

**MNIST classification**: MNIST dataset is handwritten images ($28 \times 28$) with digital numbers ranging from 0 to 9. In this experiment, we tested our method by varying MNIST dataset including original images and cluttered translated images with different size. The MNIST training set has total 60000 images, in which we use 54000 images for training and the rest 6000 for validation. Then we evaluate the performance on test dataset with 10,000 examples. As for the experimental settings, we use the similar RNN architecture (in Appendix) as RAM to make a fair comparison.

Patch feature encoding: at the location $L_t$, we extract $m$ square patches with different scales. Then we concatenate features at different scales together to get the final patch representation $x_t$ centered at location $L_t$. Note that $L_t$ is a two dimensional vector normalized in the range [-1, 1], with the image center (0, 0) and the right bottom (1, 1).

Table 1: $28 \times 28$ MNIST classification

| Model | Error rate |
|---|---|
| FC, 2 layers (256 hiddens each) | 1.69% |
| Convolutional networks, 2 layers | 1.21% |
| RAM, 6 glimpses, $8 \times 8$, 1 scale | 1.12% |
| RAM, 7 glimpses, $8 \times 8$, 1 scale | 1.07% |
| Our method (Entropy), 6 glimpses, $8 \times 8$, 1 scale | 1.05% |
| Our method (Entropy+Context), 6 glimpses, $8 \times 8$, 1 scale | 1.01% |

Since the digits in MNIST are centered, there is no need to use top-down mechanism "where to look" to initialize the search. Thus, we only leverage the bottom-up strategy as RAM, but we introduce new constrains in Eqs. 11 and 12 and want to test whether these new features get performance gain or not. The evaluation result on the MNIST test dataset is shown in Table 1. Using 6 glimpses, patch size $8 \times 8$ and 1 scale (where $m = 1$), our method with entropy beats RAM under the same parameter setting. It demonstrates that the entropy constraint in Eq. 12 is helpful. In addition, we tested whether the context constraint in Eq. 11 contributes or not, and it also shows that it can improve the classification task with more gain.

**Cluttered Translated MNIST classification**: we use the same network architecture given in the MNIST classification experiment, but evaluate on the cluttered and translated digital images. In this experiment, we use the same protocol as RAM to create the cluttered and translated MNIST data. First, we place an MNIST digit in a random location of a larger blank image $100 \times 100$ and then add $8 \times 8$ subpatches which are sampled randomly from other random MNIST digits to random locations of the image.

Table 2 shows the classification results for the models we trained on 100 by 100 Cluttered Translated MNIST with 10 pieces of clutter. The presence of clutter is kind of random noise, which makes the task much more difficult. Since our model unifies top-down and bottom-up attention, it can find the meaningful regions and avoid the negative impact from noising subpatches. In Table 2, we can see the performance of our attention model is affected less than the performance of the other models. Using the same parameters as RAM with 6 glimpses, $12 \times 12$ and 4 scales, our approach gains almost 3%, which demonstrates that our top-down and bottom-up attention unification is more powerful than RAM. By adding the entropy and global context constraints, our method gains another 3%. We can observe the similar results while using 8 glimpses.

Table 2: $100 \times 100$ Cluttered Translated MNIST classification

| Model | Error rate |
|---|---|
| FC, 2 layers (256 hiddens each) | 57.3% |
| Convolutional, 5 layers | 49.5% |
| RAM,1 glimpses, $8 \times 8$, 1 scale | 88.6% |
| RAM,1 glimpses, $12 \times 12$, 1 scale | 87.3% |
| RAM, 6 glimpses, $8 \times 8$, 1 scale | 79.3% |
| RAM, 6 glimpses, $12 \times 12$, 4 scales | 29.8% |
| RAM, 8 glimpses, $12 \times 12$, 4 scales | 24.7% |
| Our method, 6 glimpses, $12 \times 12$, 4 scale | 26.3% |
| Our method (with Entropy constraint), 6 glimpses, $12 \times 12$, 4 scale | 23.6% |
| Our method (with Entropy+Context), 6 glimpses, $12 \times 12$, 4 scale | 20.3% |
| Our method, 8 glimpses, $12 \times 12$, 4 scales | 20.7% |
| Our method (with Entropy+Context), 8 glimpses, $12 \times 12$, 4 scales | 17.3% |

**Sequential multi-digit recognition**: since the core network is based on recurrent neural networks, we can easily extend it to handle multi-digit recognition Goodfellow et al. (2014); Ba et al. (2015). Suppose that we have training images with the target $\{y_1, y_2, ..., y_T\}$, where we can use $y_1$ marks the length of sequence or use $y_T$ to indicates the end of sequence. The task is to predict the digits as it explores the local image glimpses.

Table 3: Whole sequence recognition error rates on $54 \times 54$ MNIST multi-digit SVHN dataset

| Model | Error rate |
|---|---|
| 11 layer CNN Goodfellow et al. (2014) | 3.96% |
| 10 layer CNN Ba et al. (2015) | 4.11% |
| Single DRAM Ba et al. (2015), 18 glimpses, $12 \times 12$, 2 scales | 5.1% |
| Our method (Entropy), 18 glimpses, $12 \times 12$, 2 scales | 4.58% |
| Our method (Entropy+Context), 6 glimpses, 18 glimpses, $12 \times 12$, 2 scales | 4.03% |

We take the same protocol as Goodfellow et al. (2014) on street view house number recognition task. Specifically, we crop 64 x 64 images with multi-digits at the center and then randomly sample

Table 4: The recognition accuracy on CIFAR10 dataset with PGD attack.

| Model | Accuracy |
|---|---|
| VGG16 + PGD | 47.76% |
| Our method + VGG16 + PGD | 54.3% |

to create 54x54 jittered images with similar data augmentation. We have 212243 images for training (73257 digits from training dataset and the rest from 531131 additional image), 23511 digits for validation and 26032 digits for testing. Same as DRAM Ba et al. (2015), our attention model observes 3 glimpses for each digit before making a prediction. The recurrent model keeps running until it predicts a terminal label or until the longest digit length in the dataset is reached. In the SVHN dataset, up to 5 digits can appear in an image, so it means the recurrent model will run up to 18 glimpses per image, that is 5 x 3 plus 3 glimpses for a terminal label.

To conduct a fair comparison, we use the same architecture as DRAM (refer in Appendix). The experiment results is shown in Table 3. Compared to DRAM, our model gains better result with an error rate $4.03\%$. And our result is better than 10 layer CNN, and it is also comparable to the 11 layer CNN Goodfellow et al. (2014), which uses more complex network architecture and more model parameters.

**Robust to PGD attack**: in this experiment, we analyzed and answered whether our patch-based attention model is robust to adversarial attack or not. Specifically, we will leverage projected gradient descent (PGD) to generate adversarial examples because it has been widely used to attack deep neural networks Madry et al. (2018); Pintor et al. (2021). PGD is a very powerful adversary approach, which has the following projected gradient descent on the negative loss function:

$$x^{t+1} = \Pi_{x+S}\big(x^t + \epsilon \text{sign}(\nabla_x \ell(x, y; \theta_p))\big) \tag{15}$$

where $\ell(x, y; \theta_p)$ is the loss function defined in Eq. 14, $S \subseteq \mathcal{R}^d$ formalizes the variance of manipulative space of the adversary attack, and $\theta_p$ is the set of model parameters. In order to test our model's robustness to PGD, we change equation equation 10d as follows:

$$feat = \text{VGG}(x) \tag{16}$$
$$\boldsymbol{\alpha}_t = f_y(\text{cat}(H_t, feat); \theta_y) \tag{17}$$

where VGG is the 16-layer neural networks defined in Simonyan & Zisserman (2015), and cat indicates feature concatenation. Our attention model use 6 glimpses with patch size $12 \times 12$ to make prediction on each image. As for feature extraction, we concatenate VGG16's output with $G_t$ as input in the RNN network. In the backpropagation stage, we can minimize loss in Eq. 14 and then compute gradient w.r.t. $feat$ in Eq. 16, so we can further calculate gradient w.r.t. $x$ based on chain rule. After we get $\nabla_x \ell(x, y; \theta_p)$, we can update $x$ to generate adversarial examples according to Eq. 15. In CIFAR10 classification task, we use $step = 30$ and $\epsilon = 0.3$ in Eq. 15.

Our attention model takes a patch-based approach and then sequentially decides the next location/patch to minimize the loss in Eq. 14. The result in Table 4 shows that our model is more robust to PGD attack. The intuition behind this is that our model can select better patch in each glimpse, and further it can learn some useful structure information to do better prediction.

## 6 CONCLUSION

In this paper, we propose a visual attention approach, which unifies the top-down attention mechanism with the bottom-up recurrent attention model. Considering bottom-up RAM randomly initializes the location to search target object, our top-down attention model is to answer 'where to look' the best initial location in an image. In addition, we introduce global context and entropy constraints, in order to focus attention on edge, instead of pure black/while regions. To sum, our model leverages reinforcement learning to combine top-down and bottom-up mechanism, both of which depends the reward from classification to decide next locations to attend in a sequential manner. We conduct a bunch of experiments on MNIST, CIFAR10 and SVHN datasets, and our model shows better result than RAM, DRAM and CNN (complex architecture). We also show our model is more robust than other neural network models on recognizing noise images.

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

## A    APPENDIX

**MNIST network architecture**:

Glimpse network $G_t = f_g(X_t, L_t; \theta_g)$: $f_g$ is two-layer fully connected (FC) neural networks:

$$h_g = Rect(Linear(X_t)), h_l = Rect(Linear(L_t))$$
$$G_t = Rect(Linear(h_g + h_t))$$

where $h_g$ and $h_l$ both have dimensionality 128, while the dimension of $G_t$ is 256 for all experiments.

Recurrent network: $H_t = f_h(H_{t-1}, G_t; \theta_h)$ to remember the sequence of glimpses in order to make action to get best return. $H_t = Rect(Linear(H_{t-1}) + Linear(G_t))$

Action network: $L_t = f_l(H_t; \theta_l)$ is the policy network, where its mean is the linear network over $H_t$ and its variance $\sigma$ is fixed.

**SVHN network architecture**:

the glimpse network is composed of three convolutional layers where the first layer is with filter size $5 \times 5$ and the later two layers with filter size $3 \times 3$. The number of filters in those layers was $\{64, 64, 128\}$ respectively. The recurrent model was 2-layer LSTM, with 512 hidden units in each layer.

