# OpenReview forum: "Unifying Top-down and Bottom-up for Recurrent Visual Attention"
_ICLR.cc/2022/Conference — ICLR 2022 Submitted_

### Official Review · Reviewer_ooVw · 2021-10-24

**Correctness:** 3
**Technical Novelty And Significance:** 2
**Empirical Novelty And Significance:** 2
**Recommendation:** 3
**Confidence:** 4

**Main Review:**

Strength:
1. The whole paper is well-written and well-organized.
2. The idea of combing top-down attention and bottom-up attention in the recurrent attention model is interesting and reasonable.

Weaknesses:
1. From Algorithm 1 and implementation details, the top-down model is only a one-step prediction, which makes the  whole model less convincing. In contrast, I suggest the authors can discuss multiple-step top-down predictions, and how to (dynamically) control the step of top-down prediction or trade-off the steps in two types of attention.

2. The comparisons with existing works are unfair. Since the proposed model has extra top-down steps, it is unfair to compare with the baselines at some fixed steps of glimpses. Instead, I suggest including the number of steps of the top-down steps into evaluation.

3. Motivations of the proposed context constraint are not clear. Based on my understanding, \alpha_{ty} is the normalized action probability distribution in y-th class at t-th step. Thus, based on Eq.(11), the meaning of C is the averaged hidden state of all T steps, and I am confused about the exact meaning of C.

4. The proposed entropy constraint is limited. For the MNIST-like dataset, it is reasonable to skip all black or white regions. However, for other more complex datasets or visual scenes, image patches with higher entropy seem to have no direct relation with the final prediction.


Minor:
1. Eq.(10)c seems wrong. It should be L_{t+1} instead of L_t?

Writing Suggestions:
1. It would be better to change the sequence of "context constraint" (Eq. (11)) and "entropy constraint with better exploration" (Eq. (12) - (14)) in Page 6 to make it consistent with the introduction section and article part.

**Summary Of The Paper:**

This paper extends the recurrent attention model(RAM) with another extra top-down attention. Specifically, they exploit image pyramids and Q-learning to select regions-of-interest first in the top-down attention mechanism, and then follow RAM to use policy gradient to find the patch in the bottom-up attention. Meanwhile, they also propose two loss constraints to further boost the performance of bottom-up recurrent neural networks. The proposed framework is an end-to-end framework. Experiments on three datasets (MNIST, CIFAR 10, and SVHN) have demonstrated the effectiveness of the proposed model.

**Summary Of The Review:**

The idea itself is interesting, and but I think these are several limitations of existing versions (cf. the weaknesses part), and the submission can be further improved by solving my concerns.

---

### Official Review · Reviewer_ixZU · 2021-10-30

**Correctness:** 3
**Technical Novelty And Significance:** 2
**Empirical Novelty And Significance:** 2
**Recommendation:** 3
**Confidence:** 4

**Main Review:**

### Weaknesses

- Any direct evidence to show that the initialization and attention trajectory are better than the original RAM method other than final accuracy? Can you propose some quantitative metrics and offer more visualizations? Now I do not know where the improvement comes from. Is there any possibility that the improvement simply comes from larger model capacity?
- Tab. 2: On this task, the proposed method brings significant improvement, which is amazing. But there is not any in-depth analysis about it. BTW, I think a fair comparison to CNN-based methods would be a spatial transformer network, or any other CNN with spatial attention.
- Tab. 3: Seems to me that RAM-based methods are very suitable for sequential digit recognition since they have the concept of glimpse. But it performs worse than a CNN method proposed in 2014. Why is that?

- better report run-time speed and computation cost for all the comparisons. Accuracy is not one major advantage for RAM-based methods compared to CNN-based methods (see Tab. 3)
- Tab. 2 is too difficult to read. Maybe you can report RAM and the proposed method under different settings pair by pair.

**Summary Of The Paper:**

This submission proposes to offer a better initialization by using image pyramids and Q-learning (top-down manner) for the original recurrent visual attention (RAM) model (bottom-up manner). Two new constraints are also proposed for better exploration for RAM. The proposed method has been tested on several image classification datasets, including MNIST, cluttered translated MNIST, SNHN (sequential multi-digit recognition). They also test the robustness to adversarial attack (PGD attack) on CIFAR10.

**Summary Of The Review:**

The motivation looks OK but there still remain many questions. A bunch of experiments have been conducted and some of them look very promising, but more in-depth analysis are required to address my concerns.

---

### Official Review · Reviewer_ARgD · 2021-11-03

**Correctness:** 4
**Technical Novelty And Significance:** 4
**Empirical Novelty And Significance:** 4
**Recommendation:** 6
**Confidence:** 2

**Main Review:**

Strength: The idea is interesting and new. The writing is good and easy to understand.  Sufficient experiments also demonstrate the effectiveness of the proposed model.

Weakness: The authors should give some visual examples to show the difference of the searched regions between the proposed method and other compared methods, especially the RAM model. Such visualization can also serve as qualitative evidence to support the effectiveness of the proposed method.

**Summary Of The Paper:**

In this paper, the authors propose a novel method to unify the top-down and bottom-up attention together for recurrent visual attention. They also propose two constraints in the bottom-up recurrent neural networks for better balancing the trade-off between exploration and exploitation when searching local regions.

**Summary Of The Review:**

Overall, this paper proposed an interesting idea and successfully demonstrated its effectiveness. Only a few visualizations should be done to better justify the proposed ideas.

---

### Decision · Program_Chairs · 2022-01-20

**Decision:**

Reject

**Comment:**

This submission receives mixed reviews. One reviewer leans positively while two reviewers are negative. They raise several issues upon improper evaluations, insufficient experimental analysis, baseline and sota network comparisons, presentation unclarity, and technical motivations. In the rebuttal and discussion phases, the authors do not make any response to these reviews. After checking the whole submission, the AC agrees with these two reviewers that there are several drawbacks to the aspects of the technical presentation and experimental configurations. The authors shall take these suggestions into consideration and make further improvements upon the current submission.